# Active screening significantly improves detection of chlamydia in women's health clinics in Shaoxing, China

Jieqiong Guan[1]*, Jie Feng[1], Lingying Zhu[1], Jinlong Ding[2]

1 Department of Public Health, Shaoxing Maternity and Child Health Care Hospital, Shaoxing, China,
2 Department of laboratory medicine, Shaoxing Maternity and Child Health Care Hospital, Shaoxing, China

* 450116778@qq.com

## Abstract

### Background

*Chlamydia trachomatis* (CT) infection is one of the most prevalent genital tract infections which has been reported to have adverse effects on reproductive health. However, doctors will only test for CT infection if there are associated symptoms. This study aimed to compare outcomes before and after implementing a CT screening protocol to evaluate whether to encourage more routinized CT screening.

### Methods

A hospital-based survey was conducted at Shaoxing Maternity and Child Health Care Hospital. The study was divided into two time periods, T1 (July 2021–June 2022, before screening protocol) and T2 (July 2022–June 2023, after screening protocol). The study included females aged over 16 years who agreed to CT testing at four clinics, including obstetrics, infertility, termination of pregnancy and gynecology. The nucleic acid amplification results were utilized to ascertain the presence of chlamydia infection in cervical secretions.

### Results

After active promotion of CT screening, 15,005 females were tested, with an average age of 32.2 years (SD = 8.3), which were more than double the previous number and significantly younger ($P < 0.05$). Significant differences in the number of people CT screened were observed in the obstetrics (2.5%–66.5%, $P < 0.05$) and termination of pregnancy (7.5%–45.3%, $P < 0.05$) clinics. The positive rates of CT in T1 and T2 were 6.8% (491/7230) and 5.2% (774/15005), respectively. The implementing chlamydia screening identified an additional 184 (11–195) infected pregnant women in obstetrics clinics (a similar number of outpatient visits). The age group with the highest proportion was 26–30 years (27.6%), followed by 21–25 years (21.4%) and 31–35 years (21.1%).

**Data availability statement:** All relevant data are within the paper and its Supporting Information files.

**Funding:** This study was supported by Shaoxing Health Science and Technology project (2022KY041).

**Competing interests:** The authors have declared that no competing interests exist.

## Conclusions

Active screening significantly improves the detection of chlamydia in women's health clinics. Pregnant individuals and those seeking pregnancy termination were previously overlooked in CT screening efforts but required increased attention.

## Introduction

*Chlamydia trachomatis* (CT) infection is one of the most prevalent genital tract infections and among the five sexually transmitted diseases that require prioritized attention for prevention and treatment in China. The World Health Organization aims to end the epidemic of sexually transmitted infections (STIs) by 2030 [1]. As a curable STI, CT had a global prevalence rate of 4.2% among women aged 15–49 years in 2012 [2] and an estimated rate of 3.8% in 2016 [3]. The incidence of CT infection in China has been steadily increasing over the past decade, with an average annual growth rate of 10.44% based on data from 105 sexual disease surveillance sites [4,5].

CT infection in the genital tract has been reported to have adverse effects on reproductive health, including pelvic inflammatory disease and infertility [6,7]. During pregnancy, CT infection increases the risk of stillbirth, preterm birth, ectopic pregnancy, and abortion [8,9]. Furthermore, infants born vaginally to infected mothers may suffer from low birth weight, respiratory infections, conjunctivitis, and other symptoms [10]. However, approximately 70% of infected women experience asymptomatic infection [11], resulting in a low level of willingness and awareness among patients regarding active screening.

Studies have supported the benefits of CT screening in reducing disease incidence through earlier detection and treatment [12,13]. The guidelines propose that women who are sexually active, undergoing abortion, or in the first trimester of pregnancy should undergo CT screening [14,15]. However, the evidence is insufficient to support the recommendation of routine CT screening.

Based on the StopCT Research Plan proposed by the National Center for Sexually Transmitted Disease Control [16], Shaoxing Municipal Health Commission launched a "Shaoxing comprehensive prevention and control of *Chlamydia trachomatis* infection pilot" project in July 2022 to expand the active screening of CT among patients attending reproductive health-related clinics. This study aimed to compare outcomes before and after implementing of a CT screening protocol to evaluate whether to encourage more routinized CT screening. The study could provide information for understanding the burden of CT in women and for further development of CT screening strategies in China.

## Materials and methods

### Study design and setting

The implementation of a CT screening strategy has been initiated in Shaoxing Maternity and Child Health Care Hospital since July 2022, primarily focusing on enhancing awareness among medical professionals and patients in order to

augment the CT screening rate. This study aimed to compare the CT screening and positive detection rates before and after implementing the CT screening strategy (T1: July 2021–June 2022; T2: July 2022–June 2023) in order to evaluate the effect of expanded screening. The study was conducted following the Declaration of Helsinki, and the protocol was approved by the Ethics Committee of Shaoxing Maternity and Child Health Care Hospital (Approval No. 2023001). The data used in this study were retrospective studies of medical records that were fully anonymized.

## Participants ELIGIBILITY

The study population was from the following four outpatient clinics in our hospital: (1) obstetrics clinic, which primarily focuses on providing follow-up care for pregnant women throughout their pregnancy; (2) infertility clinic, mainly for treating individuals who are unable to conceive due to various reasons; (3) termination of pregnancy clinic, which primarily caters women seeking abortion for various reasons; (4) gynecology clinic, which primarily focuses on the managing diseases related to the female reproductive system. This study included women aged 16 years and older who attended these out-patient clinics and were willing to undergo testing for *C. trachomatis* following an educational intervention by their attending physician. Individuals taking antibiotics or duplicate testing for *C. trachomatis* in the last month were excluded.

## Specimen collection and testing

The attending physician utilized sterile cotton swabs to collect cervical secretions, which were subsequently placed into sterile glass tubes and promptly dispatched to the laboratory for analysis. Real-time RCR technology was employed to detect CT DNA using a commercially available test kit. (Bioperfectus Technologies, Jiangsu, China, National instrument registration: 20183400058). Sample processing and DNA extraction were conducted in accordance with the CT kit instructions. The β-globin gene was selected as the reference gene for *Chlamydia trachomatis*, and specific primers and probes were designed for PCR amplification. The reaction program consisted of 40 cycles of incubation at 55°C for 40 seconds each, using FAM and VIC channels to automatically generate real-time amplification curves. When Ct ≤ 35.3, an S-shaped curve and exponential growth phase are observed, indicating a positive result for CT.

## Statistical analysis

The prevalence of *Chlamydia trachomatis* in various groups was determined by calculating the ratio of individuals with positive test results to the total number of individuals tested, accompanied by 95% confidence intervals. Differences between T1 and T2 were assessed with the t-test (age) and the Chi-square test (ratio). The Mantel–Haenszel test was applied to assess the presence of a linear association between CT prevalence and age. The *p*-value less than 0.05 was regarded as a statistically significant difference. All statistical analysis and charting were performed by SPSS 29.0 (SPSS Inc., Chicago) and GraphPad Prism 9.5.1 (GraphPad Software, LLC).

## Results

As displayed in Table 1, before implementing the CT screening strategy (T1), 7,230 eligible participants with an average age of 34.3 years (SD = 9.6) were included, while after implementing the CT screening strategy (T2), 15,005 eligible subjects with an average age of 32.2 years (SD = 8.3) were tested, which were more than double the number in T1 and significantly younger ($P < 0.05$). Significant differences in the number of people CT screened were observed among the four outpatient clinics when comparing T2 and T1, with significant improvements observed specifically in the obstetrics (2.5%–66.5%, $P < 0.05$) and termination of pregnancy (7.5%–45.3%, $P < 0.05$) clinics.

The positive rates of CT in T1 and T2 were 6.8% (491/7,230) and 5.2% (774/15,005), respectively. As illustrated in Fig 1, during T1, the CT detection rate was higher in termination of pregnancy (7.8%, 95% confidence interval [CI]: 5.8%–10.3%) and gynecology (7.6%, 95%CI: 6.9%–8.4%) clinics, while lower in obstetrics (6.0%, 95% CI: 3.4%–10.5%) and infertility (4.8%, 95%CI: 3.9%–5.8%) clinics. Following the expansion of CT screening, gynecology (6.7%, 95%

**Table 1. The characteristics of participants before and after implementing CT screening strategy.**

| | N1/N2 (%)[a] | | | Mean Age (standard deviation)[c] | | |
|---|---|---|---|---|---|---|
| | T1[b] | T2[b] | *P*-value | T1[b] | T2[b] | *P*-value |
| Obstetrics Clinic | 7161/182 (2.54) | 7890/5251 (66.55) | <0.001 | 29.40±4.54 | 28.99±4.32 | 0.106 |
| Infertility Clinic | 6029/2050 (34.00) | 6047/1869 (30.91) | <0.001 | 30.56±5.19 | 30.65±5.10 | 0.602 |
| Termination of Pregnancy Clinic | 7335/552 (7.52) | 6832/3098 (45.34) | <0.001 | 29.82±5.67 | 30.76±5.77 | <0.001 |
| Gynecology Clinic | 75339/4446 (5.90) | 71750/4787 (6.67) | <0.001 | 36.77±10.83 | 37.43±11.14 | 0.007 |
| Total | 95864/7230 (7.55) | 92519/15005 (16.23) | <0.001 | 34.28±9.60 | 32.23±8.28 | <0.001 |

[a]N1, the number of women attending the clinic; N2, the number of women who underwent a CT test; %, N2/N1.

[b]The T1 and T2 represent the time before and after the implementation of the CT screening strategy, respectively.

[c]the age of women who underwent a CT test.

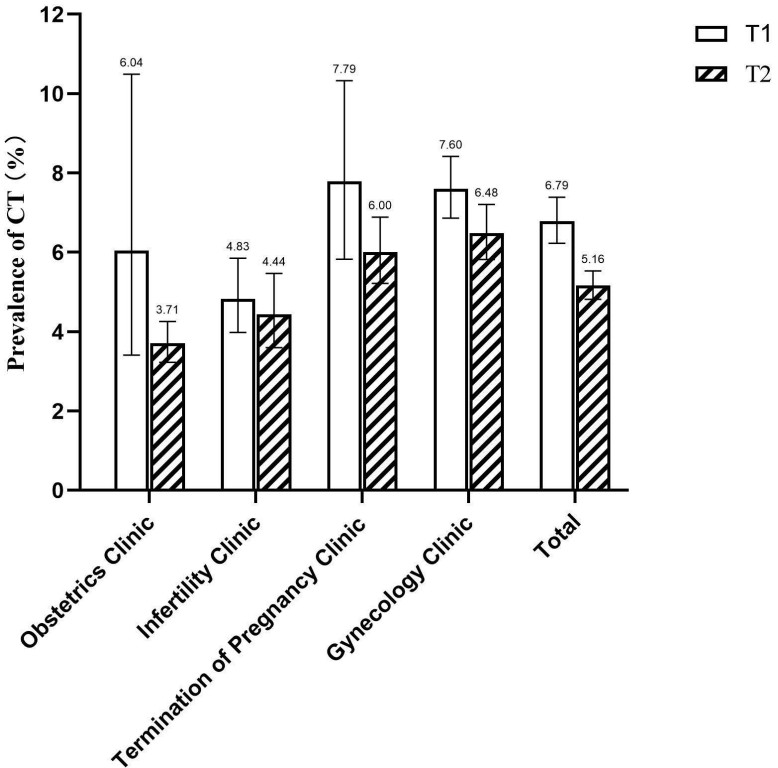

**Fig 1. The prevalence of *Chlamydia trachomatis* infection in different clinic.**

CI: 5.8%–7.2%) and termination of pregnancy (6.0%, 95% CI: 5.2%–6.9%) clinics remained the areas with the highest prevalence of CT.

As illustrated in Fig 2, although the positive detection rate of obstetric women has significantly decreased (6.0%–3.7%), implementing chlamydia screening identified an additional 184 (11–195) infected pregnant women in obstetrics clinic (a similar number of outpatient visits). Meanwhile, in the termination of pregnancy clinic, there were 143 (43–186) additional asymptomatic positive infections compared with T1. However, in the gynecology and infertility clinics, the number of chlamydia cases detected was comparable between T1 and T2.

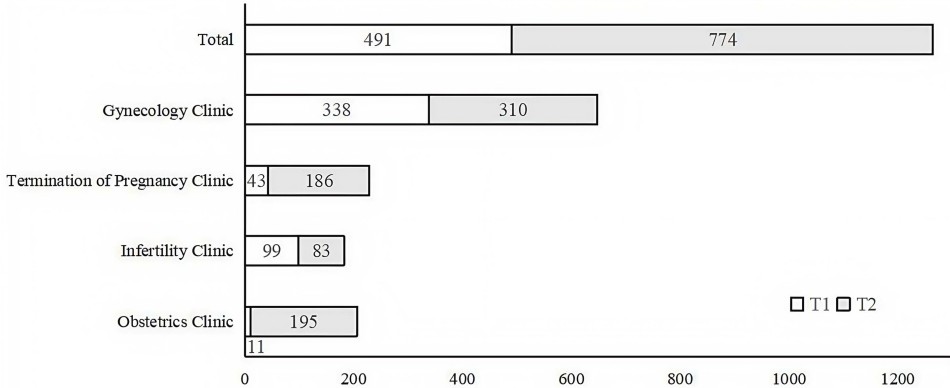

**Fig 2. The number of *Chlamydia trachomatis* positive patients in different clinic.**

The distribution of chlamydia-positive cases among different age groups is illustrated in Fig 3. The age group with the highest proportion was 26–30 years (27.6%), followed by 21–25 years (21.4%) and 31–35 years (21.1%).

## Discussion

From the 2019 Global Burden of Disease Study [17], the age-standardized incidence rate (ASR) of CT increased by 0.29% annually from 2010 to 2019 worldwide, and the ASR of CT in Southeast Asia was 1.5-fold higher than the global average. Reportedly, the average annual reported incidence rate of CT infection in Zhejiang province in the past five years was 122.50 per 100,000 women [18], while in Shaoxing City, it was 41.33 per 100,000 women [19]. The hospital serves as the primary site for CT detection in the city, accounting for a quarter of all reported cases in 2021. However, it remains unclear whether this seemingly low incidence rate of CT in Shaoxing City has been underestimated. Therefore, utilizing the StopCT Research Plan, we explored the CT prevalence in various outpatient clinics both before and after implementing expanded screening in this cross-sectional study. Finally, before and after screening, 7,230 and 15,005 people were tested, with positive rates of 6.8% and 5.2%, respectively, resulting in 491 and 774 positive cases.

These findings indicate that while CT screening has increased, the positive detection rates have not demonstrated a statistically significant increase after introducing the screening strategy protocol, which may be because doctors primarily detected

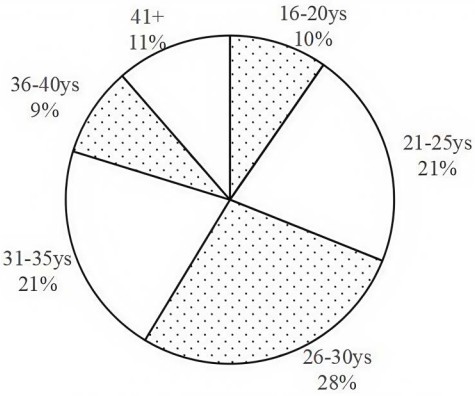

**Fig 3. The distribution of *Chlamydia trachomatis* positive cases among different age groups.**

*C. trachomatis* in symptomatic women and overlooked *C. trachomatis* detection in asymptomatic women during T1, while it is more likely to be detected in symptomatic women. Additionally, from a city-wide perspective, the significant increase in positive cases in our hospital will increase the incidence of *C. trachomatis* infection in the city, making the rate closer to the real situation.

In this hospital-based study, CT infection was found to be more common in women aged 21–35 years. The Centers for Disease Control and Prevention reported that individuals aged 15–24 years in the United States (US) represent 27% of the sexually active population, but they are responsible for approximately 50% of new STIs [20]. The Jidong Community Cohort Study involving a longitudinal serological survey of the general population in northern China revealed that at least one-quarter of the people aged 18–65 years have been infected with CT over their lifetime, while all age groups are susceptible to CT infection [21].

The detection rate of CT in the gynecology clinic in this study was 6.5%, which is consistent with the CT prevalence of 6.3% among female outpatients with genital tract infections found in a nationwide multi-center, cross-sectional study conducted by Li et al. [22] The CT detection rate in termination of pregnancy clinic at an urban public hospital in the US was slightly higher (9.6%) compared to this study (6.0%), possibly due to the younger age distribution in the US study where over half of the population was under 25 years old, while the average age in this study was 31 years old [20].

The expanded screening strategy facilitated the early detection and treatment of chlamydia in 195 pregnant women during the T2 period. Before implementing this screening, this population exhibited a lack of proactive awareness regarding CT examinations, resulting in limited testing volume and challenges in identifying infected individuals. A review of 15 studies, with 13 providing some support, suggests that prenatal screening and treatment interventions may reduce adverse pregnancy and infant outcomes for expectant mothers [8].

After publicity and education, nearly half of the women in the termination of pregnancy clinic were willing to accept the CT test, and the positive infection rate was 6.0%, which was the second among the four clinics. The CT prevalence was 10.6% in a similar study in Shenzhen [23], and the main difference may be due to the older average age in this study. It is well known that reproductive health education for the population with unwanted pregnancies is very important, and the education for CT should not be ignored.

Despite the advantage of our study in understanding the prevalence of CT infection among outpatient women in hospitals and its important implications for future prevention and control of CT, caution should be exercised when extrapolating our results to other populations due to several limitations of the present study. First, the study was conducted in a hospital setting, which may have a higher estimation of the CT prevalence, resulting from differences in sexual and healthcare-seeking behaviors among hospital outpatients compared to the general population. Furthermore, despite the hospital's active efforts to promote CT screening knowledge, there remains a reluctance among many patients to undergo such screenings. Second, the population age distribution varies across different clinics, and the observed associations are susceptible to various potential biases.

## Conclusions

In conclusion, this study identified that active screening significantly improves the detection of chlamydia in women's health clinics. Pregnant individuals and those seeking pregnancy termination were previously overlooked in CT screening efforts but required increased attention.

## Supporting information

**S1 File. Data English.**
(SAV)

## Author contributions

**Data curation:** Jie Feng, Jinlong Ding.

**Funding acquisition:** Jieqiong Guan.

**Investigation:** Jie Feng, Lingying Zhu.

**Supervision:** Lingying Zhu.

**Writing – original draft:** Jieqiong Guan.

**Writing – review & editing:** Jieqiong Guan.

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
