## [Decision Letter · Decision Letter 0]

15 Sep 2024

PONE-D-24-27749Active screening significantly improves detection of chlamydia in women' s health clinics in Shaoxing, ChinaPLOS ONE

Dear Dr. Guan,

Thank you for submitting your manuscript to PLOS ONE. After careful consideration, we feel that it does not fully meet PLOS ONE’s publication criteria as it currently stands. There is a need for you to do a major review of the manuscript. Therefore, we invite you to submit a revised version of the manuscript that addresses the points raised during the review process.

**ACADEMIC EDITOR: **

Well done for conducting this research which will add to existing scientific knowledge. After conducting a review of the manuscript, I agree with Reviewer 2 regarding the statistical analysis. The statistical analysis plan is not clear. There is a need for the authors to review the section. In addition, I will advise the authors consult a Biostatistician for guidance on appropriate statistical method and presentation of result.

The authors should please address all comments made by the reviewers with a cover letter showing step-by-step response to these comments, indicating clearly where changes were made.============================== Please submit your revised manuscript by Oct 30 2024 11:59PM. If you will need more time than this to complete your revisions, please reply to this message or contact the journal office at plosone@plos.org . Please include the following items when submitting your revised manuscript:

We look forward to receiving your revised manuscript.

Kind regards,

Ochuwa Adiketu Babah, M.Sc.PH (Epidemiology), FWACS, FMCOG

Academic Editor

PLOS ONE

Journal Requirements:

https://doi.org/10.1177/0956462417689984

In your revision ensure you cite all your sources (including your own works), and quote or rephrase any duplicated text outside the methods section. Further consideration is dependent on these concerns being addressed.

"This study was supported by Shaoxing Health Science and Technology project (2022KY041)"

5. We note that your Data Availability Statement is currently as follows: All relevant data are within the manuscript and its Supporting Information files.

Reviewers' comments:

Reviewer's Responses to Questions

**Comments to the Author**

1. Is the manuscript technically sound, and do the data support the conclusions?

Reviewer #1: Yes

Reviewer #2: Partly

2. Has the statistical analysis been performed appropriately and rigorously?

Reviewer #1: Yes

Reviewer #2: I Don't Know

3. Have the authors made all data underlying the findings in their manuscript fully available?

Reviewer #1: Yes

Reviewer #2: Yes

4. Is the manuscript presented in an intelligible fashion and written in standard English?

Reviewer #1: Yes

Reviewer #2: No

5. Review Comments to the Author

Reviewer #1: The authors have addressed an issue that continues to be topical - chlamydia trachomatis infection and its impact on reproductive health.

There are numerous typographical errors. The authors are advised to kindly address these.

Reviewer #2: I commend the authors for undertaking this study. In this research work, authors employed a hospital-based survey to compare the outcomes before and after the implementation of a chlamydia trachomatis screening protocol among women in Shaoxing Maternity and Child 38 Health Care Hospital.

Below are few comments for authors to consider:

Extensive language editing is required. This has affected the write up in several ways communicating different things to a reader, eg unclear thoughts, loss of meaning and ambiguous conclusions and to mention a few. Authors should consider engaging a certified English language expert to help in the revision of the work,

Methodology

Who was involved in taking the samples per the screening protocol? This will be helpful to readers.

The statement in lines 131-132 ‘Chi-square test was used for qualitative data and t-test was used for quantitative data’ appears unclear to communicate the actual meaning to readers. Authors should kindly clarify what they really mean by this.

Results and discussion

Findings from this research work, indicate an increase in the number of people being screened for CT in reproductive health-related clinics as illustrated in T1 (7.235) and T2 (15,017). However, there is a marginal decline in the detection of positive rates of CT in the post-intervention phase, 5.2% (774/15005), compared to before the screening program, 6.8% (491/7230). An increase in the count data in T1 screening does not necessarily suggest an increase in rates. It can be inferred from the analysis that, screening for CT has increased, but the positive detection rates have not statistically increased after the introduction of the screening strategy protocol. Authors should relook at the analysis, interpret it in that light, and subsequently discuss their findings.

6. PLOS authors have the option to publish the peer review history of their article (what does this mean? ). If published, this will include your full peer review and any attached files.

**Do you want your identity to be public for this peer review?** For information about this choice, including consent withdrawal, please see our Privacy Policy .

Reviewer #1: No

Reviewer #2: No

---

## [Author Response · Author response to Decision Letter 0]

8 Oct 2024

ACADEMIC EDITOR:

Response: We were really sorry for our careless mistakes. Thank you for your reminder. We have read the PLOS ONE's style requirements, and revised them as required.

2.We noticed you have some minor occurrence of overlapping text with the following previous publication(s), which needs to be addressed:

https://doi.org/10.1177/0956462417689984

In your revision ensure you cite all your sources (including your own works), and quote or rephrase any duplicated text outside the methods section. Further consideration is dependent on these concerns being addressed.

Response: We sincerely thank the reviewer for careful reading. As suggested by the Editors, we have checked the literature carefully, added the reference and rephrased the duplicated text. We hope that the correction will meet with approval.

3.Please provide additional details regarding participant consent. In the ethics statement in the Methods and online submission information, please ensure that you have specified (1) whether consent was informed and (2) what type you obtained (for instance, written or verbal, and if verbal, how it was documented and witnessed). If your study included minors, state whether you obtained consent from parents or guardians. If the need for consent was waived by the ethics committee, please include this information.

Response: Thank you for your reminder. We have added the more ethical details in the Methods section.

“The data used in this study were retrospective studies of medical records that were fully anonymized.”(line 70-71)

4.Thank you for stating in your Funding Statement: 

"This study was supported by Shaoxing Health Science and Technology project (2022KY041)"

Response: Thank you for your reminder. We have amended Funding Statement in our cover letter.

5.We note that your Data Availability Statement is currently as follows: All relevant data are within the manuscript and its Supporting Information files.

Response: Thank you for your reminder. We have shared the “minimal data set” for our submission.

Reviewer 1#

-The authors have addressed an issue that continues to be topical - chlamydia trachomatis infection and its impact on reproductive health.

There are numerous typographical errors. The authors are advised to kindly address these.

Response: We feel sorry for our carelessness. We have carefully checked the grammar and spelling, and we also feel great thanks for your point out.

Reviewer 2#

I commend the authors for undertaking this study. In this research work, authors employed a hospital-based survey to compare the outcomes before and after the implementation of a chlamydia trachomatis screening protocol among women in Shaoxing Maternity and Child 38 Health Care Hospital.

-Extensive language editing is required. This has affected the write up in several ways communicating different things to a reader, eg unclear thoughts, loss of meaning and ambiguous conclusions and to mention a few. Authors should consider engaging a certified English language expert to help in the revision of the work.

Response: Thank you for this suggestion. The manuscript had be reviewed and edited by the native English speakers, we hope that the new version of the article has a better reading experience.

-Methodology

Who was involved in taking the samples per the screening protocol? This will be helpful to readers.

The statement in lines 131-132 ‘Chi-square test was used for qualitative data and t-test was used for quantitative data’ appears unclear to communicate the actual meaning to readers. Authors should kindly clarify what they really mean by this.

Response: Thank the reviewer very much for pointing these out to us! We have added the screening protocol about taking the samples and the statistical analysis in the methodology.

“This study included women aged 16 years and older who attended these outpatient clinics and were willing to undergo testing for C. trachomatis following an educational intervention by their attending physician. Individuals taking antibiotics or duplicate testing for C. trachomatis in the last month were excluded.”(line 79-83)

“Differences between T1 and T2 were assessed with the t-test (age) and the Chi-square test (ratio). ”(line 100-101)

-Results and discussion

Findings from this research work, indicate an increase in the number of people being screened for CT in reproductive health-related clinics as illustrated in T1 (7.235) and T2 (15,017). However, there is a marginal decline in the detection of positive rates of CT in the post-intervention phase, 5.2% (774/15005), compared to before the screening program, 6.8% (491/7230). An increase in the count data in T1 screening does not necessarily suggest an increase in rates. It can be inferred from the analysis that, screening for CT has increased, but the positive detection rates have not statistically increased after the introduction of the screening strategy protocol. Authors should relook at the analysis, interpret it in that light, and subsequently discuss their findings.

Response: Thank you for your insightful comment and kind suggestion. We have added the suggested content to the manuscript on discussion.

“These findings indicate that while CT screening has increased, the positive detection rates have not demonstrated a statistically significant increase after introducing the screening strategy protocol, which may be because doctors primarily detected C. trachomatis in symptomatic women and overlooked C. trachomatis detection in asymptomatic women during T1, while it is more likely to be detected in symptomatic women. Additionally, from a city-wide perspective, the significant increase in positive cases in our hospital will increase the incidence of C. trachomatis infection in the city, making the rate closer to the real situation.”(line 160-167)

---

## [Decision Letter · Decision Letter 1]

7 Mar 2025

Active screening significantly improves detection of chlamydia in women' s health clinics in Shaoxing, China

PONE-D-24-27749R1

Dear Dr. Guan,

We’re pleased to inform you that your manuscript has been judged scientifically suitable for publication and will be formally accepted for publication once it meets all outstanding technical requirements.

Kind regards,

Sylvia Maria Bruisten, Ph.D

Academic Editor

PLOS ONE

Additional Editor Comments (optional):

All comments were addressed, both from the editor and from the 2 reviewers. There is only one textual remark: throughout the total text 'Chlamydia trachomatis' should be written in italics.

Reviewers' comments:

Reviewer's Responses to Questions

**Comments to the Author**

1. If the authors have adequately addressed your comments raised in a previous round of review and you feel that this manuscript is now acceptable for publication, you may indicate that here to bypass the “Comments to the Author” section, enter your conflict of interest statement in the “Confidential to Editor” section, and submit your "Accept" recommendation.

Reviewer #2: All comments have been addressed

Reviewer #3: All comments have been addressed

2. Is the manuscript technically sound, and do the data support the conclusions?

Reviewer #2: Partly

Reviewer #3: Yes

3. Has the statistical analysis been performed appropriately and rigorously?

Reviewer #2: Yes

Reviewer #3: Yes

4. Have the authors made all data underlying the findings in their manuscript fully available?

Reviewer #2: Yes

Reviewer #3: Yes

5. Is the manuscript presented in an intelligible fashion and written in standard English?

Reviewer #2: Yes

Reviewer #3: Yes

6. Review Comments to the Author

Reviewer #2: I commend the authors for the revision of the manuscript. Authors should kindly consider the following;

1. Language editing: Eg The first sentence in the abstract should read “ Chlamydia trachomatis (CT) infection, one of the most prevalent genital tract infections, has been reported to have adverse effects on reproductive health”.

The phrase “abortion women” should be revised in the entire write up.

2. Results

The statement in line 111-115 should be revised since table 1 reports on the number of people screened in each out patient clinic among the total eligible women and not chlamydia detection rates of CT.

The variable Age in Table 1 should read Mean Age (standard deviation).

Actual p values from the analysis for each tier of variables should be inculcated in the table as well for easy comprehension.

Reviewer #3: Very minimally edited, for addition of a word between long sentence parts, and/or removal of an extra word. Plus one or two minor punctuation changes (a comma), which are not essential. Otherwise, this draft is well written and summarized, now that needed text additions were made. The modified draft is uploaded. See Lines 3, 30, 47, 63, 66, 128 (Figure title), 136 (figure title).

7. PLOS authors have the option to publish the peer review history of their article (what does this mean? ). If published, this will include your full peer review and any attached files.

**Do you want your identity to be public for this peer review?** For information about this choice, including consent withdrawal, please see our Privacy Policy .

Reviewer #2: **Yes: ** Dr. Timothy Kwabena Adjei

Reviewer #3: **Yes: ** Brian L Altonen

---

## [Editor Report · Acceptance letter]

PONE-D-24-27749R1

PLOS ONE

Dear Dr. Guan,

I'm pleased to inform you that your manuscript has been deemed suitable for publication in PLOS ONE. Congratulations! Your manuscript is now being handed over to our production team.

Kind regards,

on behalf of

Dr. Sylvia Maria Bruisten

Academic Editor

PLOS ONE